# The Variation in Groundwater Microbial Communities in an Unconfined Aquifer Contaminated by Multiple Nitrogen Contamination Sources

Justin G. Morrissy [1,*], Matthew J. Currell [1,2], Suzie M. Reichman [3], Aravind Surapaneni [4,5], Mallavarapu Megharaj [6,7], Nicholas D. Crosbie [8], Daniel Hirth [9], Simon Aquilina [10], William Rajendram [11] and Andrew S. Ball [5]

1   School of Engineering, RMIT University, GPO Box 2476, Melbourne, VIC 3001, Australia; matthew.currell@rmit.edu.au
2   Water: Effective Technologies & Tools Research Centre, RMIT University, GPO Box 2476, Melbourne, VIC 3001, Australia
3   Centre for Anthropogenic Pollution Impact and Management (CAPIM), School of Biosciences, University of Melbourne, Melbourne, VIC 3010, Australia; suzie.reichman@unimelb.edu.au
4   South East Water, Frankston, VIC 3199, Australia; aravind.surapaneni@sew.com.au
5   ARC Training Centre for the Transformation of Australia's Biosolids Resource, School of Science, RMIT University, Bundoora West, VIC 3083, Australia; andy.ball@rmit.edu.au
6   Global Centre for Environmental Remediation (GCER), Faculty of Science, ATC Building, The University of Newcastle, Callaghan, NSW 2308, Australia; megh.mallavarapu@newcastle.edu.au
7   Cooperative Research Centre for Contamination Assessment and Remediation of Environment (CRC CARE), ATC Building, The University of Newcastle, Callaghan, NSW 2308, Australia
8   Melbourne Water, Docklands, VIC 3008, Australia; nick.crosbie@melbournewater.com.au
9   BlueSphere Environmental, Southbank, VIC 3006, Australia; dhirth@bluesphere-enviro.com.au
10  Gippsland Water, Traralgon, VIC 3844, Australia; simon.aquilina@gippswater.com.au
11  Greater Western Water, Sunbury, VIC 3429, Australia; william.rajendram@gww.com.au
*   Correspondence: jmorrissy.13@gmail.com

**Abstract:** Aquifers provide integral freshwater resources and host ecosystems of largely uncharacterized, truncated endemic microorganisms. In recent history, many aquifers have become increasingly contaminated from various anthropogenic sources. To better understand the impacts of nitrogen contamination on native groundwater ecosystems, 16S rRNA sequencing of the groundwater microbial communities was carried out. Samples were taken from an aquifer known to be contaminated with nitrogen from multiple sources, including fertilizers and wastewater treatment plant effluents. In total, two primary contaminants were identified: $NH_4^+$ (<0.1–3.7–26 mg $L^{-1}$ $NH_4^+$ min-median-max), and $NO_3^-$ (<0.01–18–150 mg $L^{-1}$ $NO_3^-$ min-median-max). These contaminants were found to be associated with a decrease/increase in microbial species richness within affected groundwater for $NH_4^+$/$NO_3^-$, respectively. Important phyla were identified, including Proteobacteria, which had the highest abundance within samples unaffected by $NH_4^+$ (36–81% $NH_4^+$ unaffected, 4–33% $NH_4^+$ affected), and Planctomycetes (0.05–10% $NH_4^+$ unaffected, 43–72% $NH_4^+$ affected), which had the highest abundance within the $NH_4^+$ affected samples, likely due to its ability to perform anaerobic ammonia oxidation (ANAMMOX). Planctomycetes were identified as a potential indicator for the presence of $NH_4^+$ contamination. The analysis and characterization of sequencing data alongside physicochemical data showed potential to increase the depth of our understanding of contaminant behavior and fate within a contaminated aquifer using this type of data and analysis.

**Keywords:** nitrogen contamination; groundwater; biogeochemistry; microbial biochemistry; microbial ecology; denitrification; biological N decomposition; ANAMMOX; characterization

## 1. Introduction

Aquifers serve various ecosystem functions, and are immeasurably important as resources providing potable water for humans and animals alike [1]. They also constitute a vast and poorly characterized reservoir of biological diversity, with numerous undescribed and endemic species, with significant ecological value and potential application, for example groundwater bioremediation [2–4]. Filtration and storage of water rank highly among the groundwater aquifers' ecosystem functions. Both the substrate (that isolates aquifers from the surface and constitutes the solid matrix) and the endemic ecosystems that reside within, between, and upon this substrate are integral to the effective filtration of the groundwater [3]. Their isolation from the surface (to some degree) decreases the likelihood of aquifers being contaminated by surface processes; however, it also decreases the visibility of this contamination, allowing it to remain undetected for long periods [5]. A multitude of contaminants from a range of sources globally have left a large portion of the once pristine aquifers contaminated [6]. From the 1950 onwards, aquifers and their contaminants have been chemically and hydrogeologically characterized to determine the possible effects on human health and, more recently, groundwater-dependent ecosystems (GDEs) such as springs, marshes, and swamps [5,7]. However, there has been very little characterization of the effects of contamination on the microbes that reside within the groundwater itself.

Nitrogen contamination has become ubiquitous since the Haber–Bosch process of industrial-scale nitrogen synthesis was created [6]. Nitrogen contamination can leach into the environment through point source and diffuse contamination. A typical example of a point source for nitrogen contamination is a leak from a wastewater-holding basin at a sewage/wastewater treatment plant (WWTP). Diffuse source contamination occurs when a low (but higher than natural) concentration of the contaminant is spread across a large area, such as from the use of fertilizers on croplands or cattle grazing for an extended period of time [6,8]. The effects of nitrogen contamination have been, and continue to be, documented for both individual lifeforms [9–12] and ecosystems [13]. However, there is scarce understanding of the effects of nitrogen contamination in groundwater ecosystems, particularly those residing in situ within the aquifer matrix. Characterization of the phyla in groundwater is important for several reasons; firstly, different microbes are involved in nutrient cycling within the aquifer, and different microbes can break down a variety of environmental contaminants. Additionally, many microorganisms present in groundwater will be advected into receiving water sources and detecting potentially harmful microbes early is exceptionally beneficial. Finally, as more papers characterize their sites, they begin to build a profile of the different microbial ecosystems and how they change across different habitats and contaminants; this allows us to estimate the overall health of the ecosystems in the groundwater more accurately by observing the microbial communities.

The regular sampling of bores and measurement and analysis of physicochemical properties in conjunction with a full suite of hydrochemical indicators is standard for any industrial site that has the potential to contaminate groundwater, such as WWTPs. Indicator species are used in many disciplines to detect specific environmental conditions and infer the health of an ecosystem [14]. If used correctly, indicator microbial species or phyla, when used in conjunction with physicochemical and environmental data, could add significant insight and resolution to the analysis of contaminant behavior in groundwater.

This study assessed the impact of N contamination (both $NH_4^+ \rightleftharpoons NH_3$ and $NO_3^- \rightleftharpoons NO_2^-$, henceforth referred to and $NH_4^+$ and $NO_3^-$ throughout) on the abundance and diversity of microbial communities in an aquifer contaminated from both point and diffuse sources—upstream crop farmlands, cattle grazing, and a wastewater treatment plant (WWTP). Bores with uncontaminated groundwater from this aquifer were used for an uncontaminated comparison. The primary aims of the study were to:

1.  Determine the effects of the $NO_3^-$ and $NH_4^+$ contamination on the microbial communities in the groundwater;
2.  Characterize the most abundant phyla in the groundwater and determine their importance;

3. Understand how indicators and microbial community analysis can be used in conjunction with physicochemical and environmental data to add insight and resolution to the analysis of contaminant behavior in groundwater.

## 2. Materials and methods

### 2.1. Study Area

A detailed description of the sampling site (Figure 1) can be found in Adebowale et al. [15] and McCance et al. [16]. Briefly, the site is approximately 80 km southeast of Melbourne, Australia. The site represents a typical example of a region with multiple current and historical nitrogen contamination sources. The main nitrogen groundwater plume was centered around the wastewater treatment plant (WWTP); however, due to the up-gradient intensive agriculture (market garden farms) and down-gradient cattle grazing paddocks, the normal background concentration of nitrogen, and thus the extent of influence from the WWTP versus other sources, was difficult to determine. Delineation of current and historical contamination plumes at the site are discussed in McCance, Jones, Surapaneni, and Currell [16] and McCance et al. [17]; the main plume extends from bore site RB11 across RB12, and through RB13/14 and past RB17/18 (Figure 1).

The Quaternary Bridgewater Formation was the primary aquifer of concern at this site. It is comprised of carbonate-cemented aeolian sands, is unconfined, and measures up to 100 m thick [18]. Across the region, the hydraulic conductivity of the aquifer averages approximately 20 m/day [18]. Further details about the aquifer setting and site history can be found in McCance, Jones, Surapaneni and Currell [16] and McCance, Jones, Cendón, Edwards, Surapaneni, Chadalavada, Wang, and Currell [17].

### 2.2. Sampling Timeline and Locations

For the duration of the study, 24 groundwater samples were taken in triplicate (resulting in a total of 72 subsamples) across three sampling campaigns from 10 bores. The three sampling campaigns took place in August 2018 (n = 8), November 2018 (n = 7), and May 2019 (n = 9). The locations of the sampled bores are shown in Figure 1. Further details about the bores are described in detail in Adebowale, Surapaneni, Faulkner, McCance, Wang, and Currell [15]. The groundwater flowed from southeast to northwest, starting from the south-easterly bore, and moving northwest. Bores DSE63273 and RB23 are both up-gradient from the WWTP (in terms of the regional groundwater flow direction—see McCance, Jones, Surapaneni, and Currell [16]). Bores RB10 and RB12 are adjacent to each other within the WWTP boundary; RB17/18 and RB06/07 are nested bores (one shallow, one deeper) within the Browns Road Farm down-gradient of the WWTP. Bores BS02 and BS04 are located further north along the groundwater flow direction, within the Tootgarook Swamp, which itself is within a conservation reserve. These sites currently show no indication of nitrate contamination and were considered to be uncontaminated by either the WWTP or agriculture. Further details about the groundwater flow paths can be found in McCance, Jones, Cendón, Edwards, Surapaneni, Chadalavada, Wang, and Currell [17].

### 2.3. Sampling Techniques, Technology and Guidelines

Groundwater samples were collected in accordance with EPA guidelines [19,20], using low flow sampling techniques. For monitoring bores, standing water level was first measured using a Solinst interface probe, and then continuously monitored during low-flow pumping, along with field physio-chemical parameters (electrical conductivity, oxidation-reduction potential, dissolved oxygen, temperature, and pH). These parameters were monitored using a multi-parameter field probe (YSI Pro Plus or HACH HQ40D) [15]. Samples were collected following stabilization of these parameters.

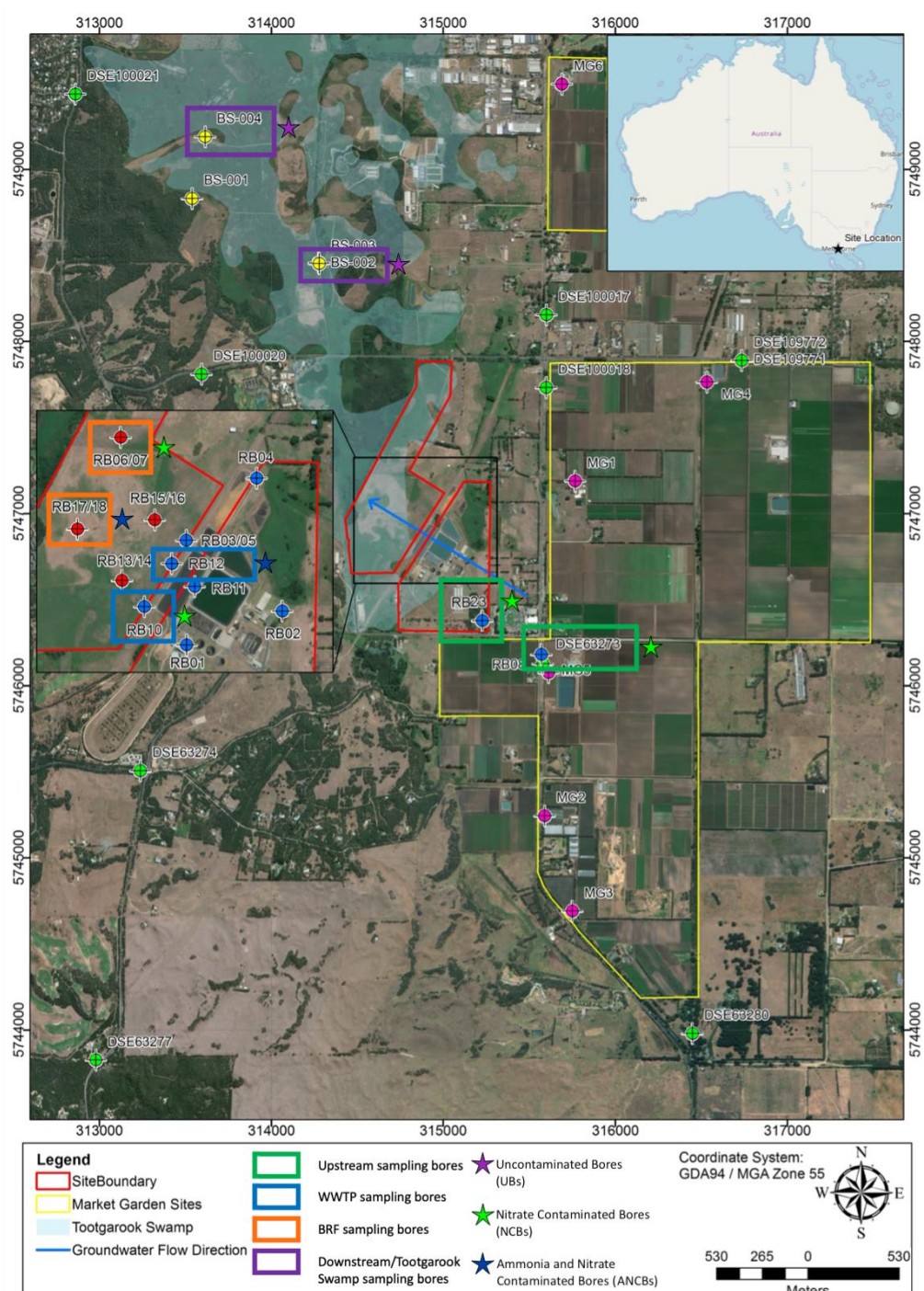

**Figure 1.** Site location and layout map. BRF = Browns Road Farm. Adapted from (Adebowale et al., 2019).

### 2.4. Major Ions and Nutrients Analysis

Samples for analysis of major ions and nutrients were collected in bottles provided by Australian Laboratory Services (ALS Laboratory) and delivered to the laboratory on the same day for analysis, using standard analytical techniques required under the National Association of Testing Authorities (NATA) accreditation. Results are reported in Table 1 (for a full list of methods, see Table S1 of Supplementary Material) [15]. Sample duplicates, triplicates, and field blanks were collected and analyzed for quality assurance; all reported data met the necessary reporting thresholds of these methods.

**Table 1.** Physiochemical and sample identification data for the upstream, WWTP bores (sludge lagoon for comparison), Browns Road Farm and the downstream Tootgarook Swamp bores. Units in (mg L$^{-1}$) unless otherwise stated.

| Relative Location | Sample | Bore Code | Date Sampled | Bore Depth | DO | EC (uS/cm) | pH | Redox Potential (mV) | Water Temp (°C) | *E. coli* (cfu/100 mL) | Alkalinity as CaCO$_3$ | HCO$_3^-$ | SO$_4$ | Anionic Strength (meq/L) | Cationic Strength (meq/L) |
|---|---|---|---|---|---|---|---|---|---|---|---|---|---|---|---|
| Upstream | AA.1 | DSE63273 | Aug-2018 | 25.08 | 4.46 | 2944 | 6.7 | 106.9 | 16.3 | 0 | 315 | 385 | 410 | 32 | 28 |
| | AB.1 | RB23 | Aug-2018 | 11.14 | 5.92 | 1487 | 7.15 | 28.9 | 15.1 | 0 | 270 | 329.4 | 24 | 17 | 14 |
| | AB.2 | RB23 | Nov-2018 | 11.18 | 5.89 | 1463 | 6.93 | 111.3 | 15.3 | 0 | 170 | 210 | 25 | 14 | 14 |
| | AB.3 | RB23 | May-2019 | 11.12 | 4.77 | 1434 | 7.27 | 33.8 | 17.0 | 0 | 170 | 210 | 45 | 13 | 12 |
| | | Ca$^{2+}$ | Na$^+$ | K$^+$ | Cl$^-$ | Mg$^{2+}$ | NH$_4^+$ as N | NO$_3^-$ as N | Total N | TOC | PO$_4^{3-}$ | TDS | Cu | Fe$^{2+}$ | Zn |
| | AA.1 | 390 | 135 | 1.2 | 230 | 41 | 0.15 | 150 | 155 | 3.7 | 0.009 | 2250 | 0.004 | 0.02 | 0.0295 |
| | AB.1 | 170 | 84 | 1 | 170 | 19 | <0.1 | 83 | 83 | 1.3 | <0.05 | 1200 | 0.003 | <0.01 | 0.006 |
| | AB.2 | 170 | 85 | <1 | 160 | 17 | <0.1 | 72 | 88 | 1.3 | <0.05 | 960 | 0.008 | <0.01 | 0.014 |
| | AB.3 | 150 | 75 | <1 | 150 | 15 | <0.1 | 64 | 65 | 2.5 | <0.005 | 1000 | 0.0092 | 0.003 | 0.0087 |
| Wastewater Treatment Plant (WWTP) | BA.1 | RB10 | Aug-2018 | 5.10 | 0.22 | 2357 | 6.68 | 32.8 | 14.8 | 2 | 610 | 744.2 | 640 | 32 | 27 |
| | BA.2 | RB10 | Nov-2018 | 5.46 | 0.51 | 2444 | 6.73 | 61.2 | 18 | 0 | 430 | 530 | 580 | 29 | 31 |
| | BA.3 | RB10 | May-2019 | 5.39 | 0.48 | 2659 | 6.72 | 21.1 | 20 | 0 | 470 | 580 | 760 | 34 | 34 |
| | BB.1 | RB12 | Aug-2018 | 5.10 | 0.15 | 2192 | 6.78 | 32.2 | 15.3 | 2 | 560 | 683.2 | 330 | 26 | 24 |
| | BB.2 | RB12 | Nov-2018 | 5.20 | 0.21 | 2641 | 6.69 | 68.7 | 18.4 | 0 | 500 | 610 | 380 | 28 | 31 |
| | BB.3 | RB12 | May-2019 | 5.14 | 0.47 | 2466 | 6.81 | 23.3 | 19.8 | 0 | 450 | 550 | 530 | 29 | 27 |
| | Sludge Lagoon | - | - | | 3.59 | 1171 | 7.51 | 44.6 | 18.6 | 4426 | 460 | 396 | 24 | - | - |
| | | Ca$^{2+}$ | Na$^+$ | K$^+$ | Cl$^-$ | Mg$^{2+}$ | NH$_4^+$ as N | NO$_3^-$ as N | Total N | TOC | PO$_4^{3-}$ | TDS | Cu | Fe$^{2+}$ | Zn |
| | BA.1 | 350 | 120 | 11 | 210 | 50 | <0.1 | 14 | 15 | 10 | <0.05 | 1800 | 0.002 | <0.01 | 0.01 |
| | BA.2 | 430 | 110 | 11 | 170 | 59 | <0.5 | 54 | 55 | 12 | <0.05 | 1700 | 0.001 | <0.01 | 0.013 |
| | BA.3 | 430 | 140 | 14 | 200 | 69 | <0.5 | 39 | 40 | 13 | <0.005 | 1900 | 0.0025 | 0.015 | 0.012 |
| | BB.1 | 230 | 120 | 40 | 190 | 48 | 26 | 40 | 66 | 10 | <0.5 | 1400 | 0.004 | <0.01 | 0.009 |
| | BB.2 | 320 | 160 | 40 | 210 | 61 | 13 | 63 | 76 | 12 | 0.05 | 1600 | 0.002 | <0.01 | 0.015 |
| | BB.3 | 310 | 140 | 31 | 170 | 54 | 5.5 | 54 | 60 | 11 | <0.005 | 1600 | 0.0034 | 0.009 | 0.0086 |
| | Sludge Lagoon | 37 | 82 | 44 | 107 | 20 | 54 | 0.17 | 57 | 27 | 70 | 493 | 0.11 | 3.5 | 0.25 |

**Table 1.** *Cont.*

| Relative Location | Sample | Bore Code | Date Sampled | Bore Depth | DO | EC (uS/cm) | pH | Redox Potential (mV) | Water Temp (°C) | *E. coli* (cfu/100 mL) | Alkalinity as $CaCO_3$ | $HCO_3^-$ | $SO_4$ | Anionic Strength (meq/L) | Cationic Strength (meq/L) |
|---|---|---|---|---|---|---|---|---|---|---|---|---|---|---|---|
| Browns Road Farm | CA.1 | RB17 | Aug-2018 | 4.74 | 0.24 | 1486 | 6.92 | 30.5 | 14.4 | 0 | 350 | 430 | 220 | 17 | 18 |
| | CA.2 | RB17 | Nov-2018 | 4.74 | 0.2 | 1723 | 6.84 | 77.2 | 15.1 | 0 | 370 | 450 | 330 | 20 | 18 |
| | CA.3 | RB17 | May20-19 | 4.68 | 0.64 | 1682 | 7.01 | 29.1 | 17.1 | 1 | 360 | 440 | 250 | 18 | 18 |
| | CB.1 | RB18 | Aug-2018 | 10.80 | 0.17 | 1381 | 6.90 | 20.7 | 15.4 | 0 | 350 | 430 | 180 | 16 | 15 |
| | CB.2 | RB18 | Nov-2018 | 10.80 | 0.19 | 1309 | 6.87 | 79.9 | 15.5 | 0 | 360 | 440 | 130 | 14 | 13 |
| | CB.3 | RB18 | May-2019 | 10.77 | 0.56 | 1480 | 7.24 | 29.0 | 16.0 | 0 | 370 | 450 | 190 | 16 | 13 |
| | CC.1 | RB06 | Aug-2018 | 9.15 | 0.12 | 2633 | 6.79 | 33.4 | 14.7 | 0 | 330 | 400 | 1100 | 37 | 37 |
| | CC.2 | RB06 | May-2019 | 9.01 | 0.61 | 1915 | 7.19 | 35.9 | 15.3 | 0 | 270 | 330 | 490 | 22 | 18 |
| | CD.1 | RB07 | Aug-2018 | 4.65 | 0.22 | 3342 | 6.78 | 24.0 | 13.6 | 0 | 410 | 490 | 1500 | 48 | 36 |
| | CD.2 | RB07 | May-2019 | 4.64 | 0.76 | 3503 | 6.73 | −8.4 | 15.8 | 0 | 420 | 510 | 1500 | 48 | 36 |

| | $Ca^{2+}$ | $Na^+$ | $K^+$ | $Cl^-$ | $Mg^{2+}$ | $NH_4^+$ as N | $NO_3^-$ as N | Total N | TOC | $PO_4^{3-}$ | TDS | Cu | $Fe^{2+}$ | Zn |
|---|---|---|---|---|---|---|---|---|---|---|---|---|---|---|
| CA.1 | 200 | 120 | <1 | 150 | 28 | <0.1 | 12 | 13 | 2.4 | <0.05 | 1000 | 0.003 | <0.01 | 0.011 |
| CA.2 | 200 | 120 | <1 | 160 | 32 | <0.1 | 15 | 16 | 7.1 | <0.05 | 1200 | <0.001 | <0.01 | 0.006 |
| CA.3 | 200 | 130 | 0.8 | 160 | 27 | <0.1 | 14 | 16 | 1.6 | <0.005 | 1000 | 0.001 | 0.003 | 0.0043 |
| CB.1 | 170 | 100 | 3 | 140 | 25 | 4.1 | 18 | 20 | 2.3 | <0.05 | 840 | 0.003 | <0.01 | 0.007 |
| CB.2 | 140 | 100 | 3 | 110 | 22 | 3.3 | 15 | 18 | 6.1 | <0.05 | 830 | <0.001 | <0.01 | 0.004 |
| CB.3 | 130 | 110 | 3.5 | 140 | 22 | 4.6 | 11 | 20 | 1.3 | 0.006 | 790 | 0.0012 | 0.003 | 0.0052 |
| CC.1 | 480 | 170 | <1 | 250 | 63 | <0.1 | 7.5 | 8.2 | 8.1 | <0.05 | 2100 | 0.002 | <0.01 | 0.032 |
| CC.2 | 210 | 110 | 0.8 | 180 | 34 | <0.1 | 18 | 20 | 4.4 | <0.005 | 1200 | <0.001 | <0.01 | 0.005 |
| CD.1 | 570 | 210 | 7 | 320 | 81 | 0.1 | <0.01 | 0.9 | 16.0 | 0.08 | 2800 | 0.002 | 0.9 | 0.01 |
| CD.2 | 470 | 160 | 1 | 300 | 70 | <0.1 | 0.22 | 1.2 | 13.0 | <0.005 | 2700 | 0.001 | 1.2 | 0.01 |

| Relative Location | Sample | Bore Code | Date Sampled | Bore Depth | DO | EC (uS/cm) | pH | Redox Potential (mV) | Water Temp (°C) | *E. coli* (cfu/100 mL) | Alkalinity as $CaCO_3$ | $HCO_3^-$ | $SO_4$ | Anionic Strength (meq/L) | Cationic Strength (meq/L) |
|---|---|---|---|---|---|---|---|---|---|---|---|---|---|---|---|
| Downstream /Tootgarook Swamp | DA.1 | BS-002 | Nov-2018 | 26.35 | 0.24 | 940 | 7.43 | 5.5 | 15.5 | 0 | 240 | 300 | 8 | 9 | 9 |
| | DA.2 | BS-002 | May-2019 | 26.35 | 0.74 | 1004 | 7.37 | −59.6 | 15.3 | 0 | 250 | 300 | 7 | 10 | 8 |
| | DB.1 | BS-004 | Nov-2018 | 5.49 | 0.17 | 718 | 7.44 | −106.9 | 15.1 | 0 | 210 | 250 | 26 | 7 | 7 |
| | DB.2 | BS-004 | May-2019 | 5.49 | 0.69 | 776 | 7.77 | −83.5 | 14.3 | <10 | 210 | 260 | 34 | 8 | 7 |

| | $Ca^{2+}$ | $Na^+$ | $K^+$ | $Cl^-$ | $Mg^{2+}$ | $NH_4^+$ as N | $NO_3^-$ as N | Total N | TOC | $PO_4^{3-}$ | TDS | Cu | $Fe^{2+}$ | Zn |
|---|---|---|---|---|---|---|---|---|---|---|---|---|---|---|
| DA.1 | 84 | 82 | 1 | 160 | 11 | 0.1 | <0.01 | 0.2 | 2.6 | <0.05 | 440 | <0.001 | 0.13 | 0.008 |
| DA.2 | 83 | 78 | 1.3 | 170 | 11 | <0.1 | 0.01 | 0.07 | <0.5 | <0.005 | 480 | 0.0003 | 0.096 | 0.0026 |
| DB.1 | 47 | 52 | 4 | 93 | 25 | 0.2 | <0.01 | 0.3 | 3.1 | <0.05 | 300 | <0.001 | 1.4 | 0.003 |
| DB.2 | 51 | 49 | 4 | 100 | 28 | <2 | <0.2 | 0.66 | 2.7 | 0.05 | 360 | <0.0001 | 0.57 | 0.0007 |

### 2.5. Microbial Analysis

2.5.1. Sampling

Samples for microbial analysis were collected in sterilized 1 L round borosilicate amber glass laboratory bottles. Bottles were sterilized in an autoclave at a standard temperature and pressure before being sealed and transported to the sampling location. After collection, samples were transported back to the lab and filtered using a Microfil mixed cellulose esters 0.22 μm white gridded, sterile filter within a sterile filter apparatus. Filtered samples were then stored at −20 °C in a freezer within 24 h of collection.

2.5.2. DNA Extraction and Sequencing

Once the samples from all sampling rounds were processed, they were removed from the freezer, and the DNeasy PowerWater Kit was used to extract the DNA from the filtered samples. After extraction, samples were processed for 16S rRNA sequencing using a Nextera® XT Index Kit (Illumina, San Diego, CA, USA), as outlined in the 16S Metagenomic Sequencing Library Preparation guide provided by Illumina. The DNA from the library was quantified using a Qubits 2.0 Fluorometer (Life Technologies, Carlsbad, CA, USA) and a 2100 Bioanalyzer (Agilent Technologies, Santa Clara, CA, USA). The yield and purity of the DNA was standardized to the level required by the Illumina DNA Prep Reference Guide (>100 ng DNA, no organic contaminants and <1 mM EDTA). The samples were pooled and run in a MiSeq platform (Illumina, San Diego, CA, USA) at the School of Science, RMIT University [21].

2.5.3. File Preparation and Data Analysis

Following sequencing, raw fastq files were copied from the MiSeq machine and loaded into R studio along with all of the collected environmental data (including the major ions and nutrients), where all further processing and analysis was completed. Raw files were trimmed, chimeric reads were removed, and reads were grouped into operational taxonomic units (OTUs), and assigned taxonomic classifications. This was completed using the Applecorn script (https://github.com/MonashBioinformaticsPlatform/applecorn Last accessed 10 February 2021), and taxonomic classifications used the silva (132) database for reference. Alpha diversity metrics, bar graphs, and constrained correspondence analysis (CCA) were produced using several R packages, including vegan and ggplot; for a full list of packages, see Supplementary Table S2. *p* values showing the significance of correlations between physicochemical properties and phyla were obtained using a generalized linear model in the edgeR package in R, and adjusted using Bonferroni correction (with a confidence interval of 99%) to account for false positives. Bonferroni correction has likely resulted in far more false negatives due to the harshness of this *p* value correction (which reduced 427 significant results across the dataset to 73); however, given the size of the data, this allowed us to focus on only the most significant results.

## 3. Results and Discussion

### 3.1. Physicochemical Parameters

The primary contaminant of concern originating from both the diffuse source upstream market gardens (MG) and within the WWTP was nitrogen, in the form of nitrate ($NO_3^-$) and ammonia ($NH_4^+$).

Nitrate was present in concentrations of 64–150 mg $L^{-1}$ in the upstream (US) bores. Concentrations of ammonia and nitrate ranged from <0.1 mg $L^{-1}$ to 26 mg $L^{-1}$ and 14 mg $L^{-1}$ to 63 mg $L^{-1}$, respectively, in bores on the WWTP site, and decreased moving from the WWTP through the Browns Road Farm (Table 1). The Tootgarook Swamp bores, which are down-gradient of the WWTP and Browns Road Farm, remained unaffected by N contamination. The high $NO_3^-$ concentrations seen in the upstream bores can be explained, at least in part, by the high dissolved oxygen (DO) concentrations (>4 mg $L^{-1}$ $O_2$) that were present in these bores that were likely preventing denitrification from occurring [2,22]. It is unclear how much the WWTP and Browns Road Farm contributed to the larger plume

originating from the direction of the Market Gardens and covering the majority of both the WWTP and the BRF; however, contamination originating from the WWTP appears to input high concentrations of $NH_4^+$ into groundwater relative to $NO_3^-$, in contrast to the Market Gardens bores (where $NO_3^-$ was the dominant contaminant).

Water sampled from bore RB07 displayed much lower $NO_3^-$ concentrations than its nested bore (RB06), which screened the aquifer deeper than RB07. It is unclear whether this was due to the bore being on the edge of the $NO_3^-$ plume, with limited contamination, or removal by denitrification. Higher concentrations of $Fe^{2+}$ and total organic carbon (TOC) in this bore (relative to others sampled) may have provided electron donors for heterotrophic and chemoautotrophic denitrification reactions [2].

In contrast to the $NO_3^-$ plume, there was stronger evidence that the $NH_4^+$ contamination plume originated from point source contamination around RB11, likely from current and historical leaks in the effluent holding basin and sludge lagoons in the immediate area. The high $NH_4^+$ present in the sludge lagoon is possibly due to insufficient mixing resulting in anaerobic digestion or the dewatering process concentrating the $NH_4^+$. This contamination plume extends through RB12 with $NH_4^+$ concentrations of 5.5–26 mg $L^{-1}$ and past RB18, with concentrations of 3.3–4.6 mg $L^{-1}$ [16,17]. Despite RB17 residing within the $NH_4^+$ impacted area (nested with RB18 at a shallower depth), it showed no $NH_4^+$ contamination; this may be due to the $NH_4^+$ being nitrified or otherwise utilized by the microbiota in the shallower part of the aquifer. This could have occurred due to variations in the chemical and/or microbial community between RB17 and RB18.

There was negligible presence of $NH_4^+$ in the upstream bores despite nitrogen fertilizers containing equal ratios of $NO_3^-$ and $NH_4^+$. Adsorption, assimilation, and nitrification likely contribute to the lack of $NH_4^+$ in these bores (within market garden areas) due to three factors. Firstly, $NH_4^+$ is positively charged and adsorbs to the predominantly negatively charged, acidic substrates which make most soils and aquifers, giving it less mobility than $NO_3^-$ [2,23,24]. Secondly, in acidic soils and in the presence of $K^+$, $NH_4^+$ is slightly preferred over $NO_3^-$ [25]. Lastly, in oxygen-rich soils, which is typical of surface soils where fertilizers are applied, $NH_4^+$ is readily oxidized into $NO_3^-$. The combination of these three factors means that $NH_4^+$ is more likely to remain in the location where it was deposited and be assimilated by crops and/or oxidized into $NO_3^-$. In addition, adsorption and nitrification are assisted in this case by the diffuse nature of the pollution, providing a large area over which DO can be replenished.

Upstream bores in this study (DSE63273 and RB23) do not act as typical 'background bores' due to the diffuse contamination affecting the region. To combat this and to obtain samples that resemble (as close as is practical) a 'pristine' or 'uncontaminated' groundwater sample for analysis of microbial community, the Tootgarook Swamp bores were chosen. These bores exhibit no evidence of N contamination (Table 1), but they are relatively reducing (likely due to their position towards the end of the regional flow path), so this should be taken into account when assessing their ability to reflect 'background' conditions in the aquifer.

*3.2. Microbial Community Structure and Variation*

3.2.1. Diversity Metrics

For the sake of clarity and following Spellerberg and Fedor [26], in this paper, 'richness' refers to the number of species, OTUs, or phyla in a sample/area; 'diversity' refers to the diversity indices such as the Gini–Simpson index (which more heavily weights abundant species) and the Shannon–Wiener index (which more heavily weights rare species); and 'evenness' describes the degree to which abundances are divided equitably between species represented by Pielou's Evenness [27,28].

If the Tootgarook Swamp bores were the 'background' for diversity metrics, then according to the alpha diversity analysis, the Gini–Simpson index averages were higher in the bores contaminated by both $NH_4^+$ and $NO_3^-$. In addition to this, the same bores also displayed lower average Shannon–Wiener Index and Pielou's Evenness scores. This

indicates that the $NH_4^+$ and $NO_3^-$ contaminated bores may have had higher diversity in more abundant species, lower diversity in rare species, and lower evenness. Despite RB17 not presenting with elevated $NH_4^+$ concentrations, this sample followed the trend of the other bores within the $NH_4^+$ contamination plume. This suggests that despite the lower $NH_4^+$ concentrations, RB17 was likely still being affected by the contamination plume. This could indicate periodic or consistent $NH_4^+$ contamination that was consumed before physicochemical testing detects it.

In the bores only affected by $NO_3^-$ contamination, no difference from the Tootgarook Swamp bores in the average Gini–Simpson index were observed; however, higher average Shannon–Wiener Index and slightly increased average Pielou's Evenness scores than the Tootgarook Swamp bores were observed. The abundance estimators ACE and Chao1 showed no notable trends (Table 2).

From the physicochemical and diversity analysis, we can thus broadly identify three contamination conditions:

1. Bores uncontaminated by nitrogen, BS-002 and BS-004 (henceforth UBs);
2. $NO_3^-$ contaminated bores DSE63273, RB23, RB10 and RB6/7 (henceforth NCBs);
3. $NH_4^+$ and $NO_3^-$ contaminated bores RB12 and RB17/18 (henceforth ANCBs).

Statistical analysis using ANOVA showed that the Gini–Simpson index, Shannon–Wiener index, and Pielou's Evenness all showed an overall significant difference between the three contaminant conditions (F value: 0.00406, 0.00656 and 0.00546 respectively) (Supplementary Table S5). Post-hoc analysis with a Turkey's test showed that the UBs showed no significant difference from the NCBs in any of the diversity or evenness measures ($p > 0.05$) (Supplementary Table S5). The Ubs were significantly lower than the ANCBs in the Gini–Simpson index, but not significantly different in the Shannon–Wiener index or Pielou's Evenness (Supplementary Table S5). The NCBs were also significantly lower than the ANCBs in the Gini–Simpson index, but significantly higher in the Shannon–Wiener index and Pielou's Evenness (Supplementary Table S5). Chao1 and ACE showed no overall significance across the contaminant conditions (Supplementary Table S5). These statistical results indicate that the three contamination conditions are on a spectrum with NCBs on one end, ANCBs on the other, and UBs somewhere in the middle, likely slightly closer to the NCB end. This explains why the NCBs and ANCBs are so different from each other, but both are still similar to the UBs.

### 3.2.2. Spatial Variation within the Microbial Community

Within the UBs, NCBs and ANCBs there were 11, 17, and 12 phyla, respectively, that on average accounted for more than 1% of the relative abundance of the sample, and a total of 59, 63, and 55 phyla present overall. In addition, the top 20 species within UBs, NCBs, and ANCBs contributed 98%, 97%, and 99% of the total abundance, respectively. There were two phyla (Asgardaeota and Parabasilia) and 886 species endemic to NCBs, 0 phyla and 338 species endemic to ANCBs, and three phyla (Cloacimonetes, Margulisbacteria, and Modulibacteria) and 348 species endemic to UBs. Phyla and species present in some bores and not others were likely a result of local redox and geochemical conditions, though a large number of species localized to a specific contaminant may be an indicator of invasive species. Potentially invasive and opportunistic species in the NCBs and ANCBs is reasonable, as nitrogen is typically a limiting nutrient in groundwater; the introduction of nitrogen would make the environment more desirable for organisms that are usually incapable of surviving the typically limiting groundwater environment. It is likely that there are more endemic species in the NCBs vs. the ANCBs, since $NH_4^+$ is more toxic than $NO_3^-$, making the environment less desirable when $NH_4^+$ is present. The above observations also reinforce the hypothesis that ANCBs have lower overall richness and NCBs higher overall richness. The ANCBs also had fewer phyla than the UBs, which may be due to the loss of native species, likely through the out-competition of more $NH_4^+$ tolerant species.

**Table 2.** Alpha diversity, abundance estimators and evenness measures of the microbial communities for the four sampling locations.

| Sample Location | Sample | Bore Code | Ammonia as N (mg L$^{-1}$) | Number of Replicates | Non-Chimeric Sequences | OTUs | ACE | Chao1 | Gini-Simpson | Shannon-Wiener | Pielou's Evenness |
|---|---|---|---|---|---|---|---|---|---|---|---|
| | AA.1 | DSE63277 | 0.15 | 3 | 6403 | 1289 | 1429 | 1524 | 0.01 | 5.41 | 0.76 |
| | AB.1 | RB23 | <0.1 | 3 | 12,150 | 1153 | 1227 | 1261 | 0.01 | 5.72 | 0.81 |
| Upstream | AB.2 | RB23 | <0.1 | 3 | 26,674 | 1189 | 1251 | 1260 | 0.03 | 5.11 | 0.72 |
| | AB.3 | RB23 | <0.1 | 3 | 22,156 | 1263 | 1309 | 1404 | 0.04 | 4.60 | 0.64 |
| | **Average** | | **0.1** | **3** | **16,846** | **1224** | **1304** | **1362** | **0.02** | **5.21** | **0.73** |
| | BA.1 | RB10 | <0.1 | 3 | 12,535 | 1680 | 1880 | 3333 | 0.02 | 5.62 | 0.76 |
| | BA.2 | RB10 | <0.5 | 3 | 1584 | 263 | 291 | 779 | 0.11 | 3.56 | 0.64 |
| | BA.3 | RB10 | <0.5 | 3 | 21,782 | 2590 | 2717 | 2742 | 0.01 | 6.58 | 0.84 |
| WWTP | **Average** | | **<0.5** | **3** | **11,967** | **1511** | **1629** | **2285** | **0.04** | **5.25** | **0.74** |
| | BB.1 | RB12 | 26 | 3 | 7353 | 1705 | 1907 | 2069 | 0.13 | 3.58 | 0.48 |
| | BB.2 | RB12 | 13 | 3 | 23,760 | 2391 | 2551 | 2669 | 0.11 | 3.84 | 0.49 |
| | BB.3 | RB12 | 5.5 | 3 | 18,086 | 2170 | 2356 | 5870 | 0.04 | 5.35 | 0.70 |
| | **Average** | | **14.8** | **3** | **16,400** | **2089** | **2271** | **3536** | **0.09** | **4.26** | **0.56** |
| | CA.1 | RB17 | <0.1 | 2 | 21,920 | 1643 | 1790 | 1772 | 0.10 | 4.20 | 0.57 |
| | CA.2 | RB17 | <0.1 | 2 | 17,700 | 1599 | 1758 | 1748 | 0.09 | 3.98 | 0.54 |
| | CA.3 | RB17 | <0.1 | 3 | 1542 | 415 | 432 | 574 | 0.02 | 4.75 | 0.79 |
| | **Average** | | **<0.1** | **2** | **13,721** | **1219** | **1327** | **1365** | **0.07** | **4.31** | **0.63** |
| | CB.1 | RB18 | 4.1 | 3 | 30,838 | 1591 | 1703 | 1916 | 0.10 | 3.97 | 0.54 |
| | CB.2 | RB18 | 3.3 | 3 | 17,382 | 1374 | 1459 | 1652 | 0.10 | 3.92 | 0.54 |
| Browns Road Farm | CB.3 | RB18 | 4.6 | 3 | 1691 | 398 | 409 | 593 | 0.02 | 4.91 | 0.82 |
| | **Average** | | **4.0** | **3** | **16,637** | **1121** | **1191** | **1387** | **0.07** | **4.26** | **0.63** |
| | CC.1 | RB06 | <0.1 | 3 | 1964 | 758 | 837 | 1664 | 0.01 | 5.55 | 0.84 |
| | CC.2 | RB06 | <0.1 | 3 | 2284 | 556 | 578 | 858 | 0.01 | 5.19 | 0.82 |
| | CD.1 | RB07 | 0.1 | 3 | 15,628 | 2906 | 3099 | 3345 | 0.01 | 6.45 | 0.81 |
| | CD.2 | RB07 | <0.1 | 3 | 36,019 | 2894 | 3075 | 6421 | 0.01 | 6.34 | 0.80 |
| | **Average** | | **0.1** | **3** | **13,974** | **1779** | **1897** | **3072** | **0.01** | **5.88** | **0.82** |

**Table 2.** *Cont.*

| Sample Location | Sample | Bore Code | Ammonia as N (mg L$^{-1}$) | Number of Replicates | Non-Chimeric Sequences | OTUs | ACE | Chao1 | Gini-Simpson | Shannon-Wiener | Pielou's Evenness |
|---|---|---|---|---|---|---|---|---|---|---|---|
| Tootgarook Swamp | DA.1 | BS-002 | 0.1 | 3 | 9588 | 940 | 994 | 1039 | 0.03 | 4.92 | 0.72 |
| | DA.2 | BS-002 | <0.1 | 3 | 33,592 | 1534 | 1589 | 1614 | 0.05 | 4.65 | 0.63 |
| | DB.1 | BS-004 | 0.2 | 3 | 13,032 | 1882 | 2066 | 2090 | 0.01 | 6.03 | 0.80 |
| | DB.2 | BS-004 | <2 | 3 | 1809 | 379 | 389 | 610 | 0.02 | 4.87 | 0.82 |
| | **Average** | | **0.2** | **3** | **14,505** | **1184** | **1260** | **1338** | **0.03** | **5.12** | **0.74** |

Across the three contamination conditions, 13 phyla were identified that account for the majority of the abundance and variation (Figure 2). Cumulatively, the most abundant phylum was Proteobacteria; within the UBs Proteobacteria had an average relative abundance of 68.7%. Abundance reduced to 58.0% in the NCBs and 30.5% in the ANCBs (Figure 2). Proteobacteria consists of a multitude of species important to various ecosystem functions, including the nitrogen and carbon cycle [29]. Within the Proteobacteria phylum, species involved in the processing of nitrogen are typically identified by their prefix. The analyzed samples contain several families and genera that process nitrogen. For example, the prefix *Nitroso* in the *Nitrosomonadaceae* family indicates the ability to oxidize ammonia; the prefix *nitro* in the *Nitrotoga* candidate genus indicates the ability to oxidize nitrite, and the prefix *denitr* in the *Denitratisoma* genus indicates the ability to denitrify [30–32]. Additionally, there were proteobacteria capable of iron oxidation, such as the *Gallionella* genus and several families that contain pathogenic species such as the *Enterobacteriaceae* and *Pseudomonadaceae* families [33,34]. *Enterobacteriaceae* and *Pseudomonadaceae* both contain species of opportunistic human pathogens such as *E. coli* and *Pseudomonas aeruginosa*, which is a multi-drug resistant human pathogen. The second most abundant phylum, planctomycetes, is the only phylum known to contain species capable of performing anaerobic ammonia oxidation (ANAMMOX), and had an average relative abundance of 0.8% in the UBs, 2.5% in the NCBs and 42.4% in the ANCBs (Figure 2). Within the analyzed samples, there were several genera known to perform ANAMMOX, such as *Candidatus Brocadia*, *Candidatus Kuenenia* and *Candidatus Scalindua*. Additionally, these genera are known to be anaerobic, slow growing species involved in the nitrogen, carbon and sulfur cycles [35]. The ANAMMOX species (*Candidatus Brocadia*) are particularly important, as they can simultaneously transform $NH_4^+$ and $NO_2^-$ into $N_2$, which is a much more efficient process than the transformation from $NH_4^+$ to $NO_3^-$ to $N_2$ [36]. It is also highly likely that high abundance of the ANAMMOX species is closely linked with high concentrations of $NH_4^+$ in the groundwater.

The next 10 most abundant phyla are presented in their order of abundance in the UBs: The Bacteroidetes phylum is a phenotypically diverse group, and had an average relative abundance of 4.2% in the UBs, 3.9% in the NCBs, and 1.9% in the ANCBs (Figure 2). Within the analyzed samples, the Bacteroidetes phylum contained genera, such as the *Flavobacterium* genus which contains species capable of denitrification, and species capable of nitrate reduction, as well as potentially pathogenic genera, such as *Capnocytophaga*. In addition, these samples also contained the *Pedobacter* genus; strains within this genus can degrade a range of organic compounds, such as diesel.

The Epsilonbacteraeota phylum had an average relative abundance of 3.3% in the UBs, 1.0% in the NCBs, and 0.9% in the ANCBs (Figure 2). Epsilonbacteraeota is a recently proposed reclassification of the *Epsilonproteobacteria* class and the *Desulfurellales* order within the proteobacteria phylum as a result of 16S and 23S rRNA evidence [37]. Within the samples most species of Epsilonbacteraeota were microaerophilic chemoorganotrophs, and some species within the *Sulfurospirillum* genus are reported to be able to reduce $NO_3^-$ and $NO_2^-$ into $NH_4^+$ [38]. This phylum also included several pathogenic species within, but not restricted to, the *Campylobacter* genus [39]. *Campylobacter* is a typically non-lethal infection causing diarrhea and gastrointestinal distress [39].

The Actinobacteria phylum is one of the major phyla in the bacterial domain and had an average relative abundance of 2.7% in the UBs, 3.0% in the NCBs, and 1.5% in the ANCBs (Figure 2). Actinobacteria is known for its prominence in producing important molecules for various medical uses such as antibiotics [40,41]. In addition, Actinobacteria cause much of the antibiotic resistance present in environmental bacteria [41]. Within the samples analyzed nitrate and nitrite reduction was present in multiple anaerobic and facultatively anaerobic genus, such as *Cellulomonas* and *Oerskovia* [42,43]. Species within the Actinobacteria phylum can degrade a wide variety of organic compounds; several species inhabit the human gastrointestinal tract, and several species are pathogenic [44].

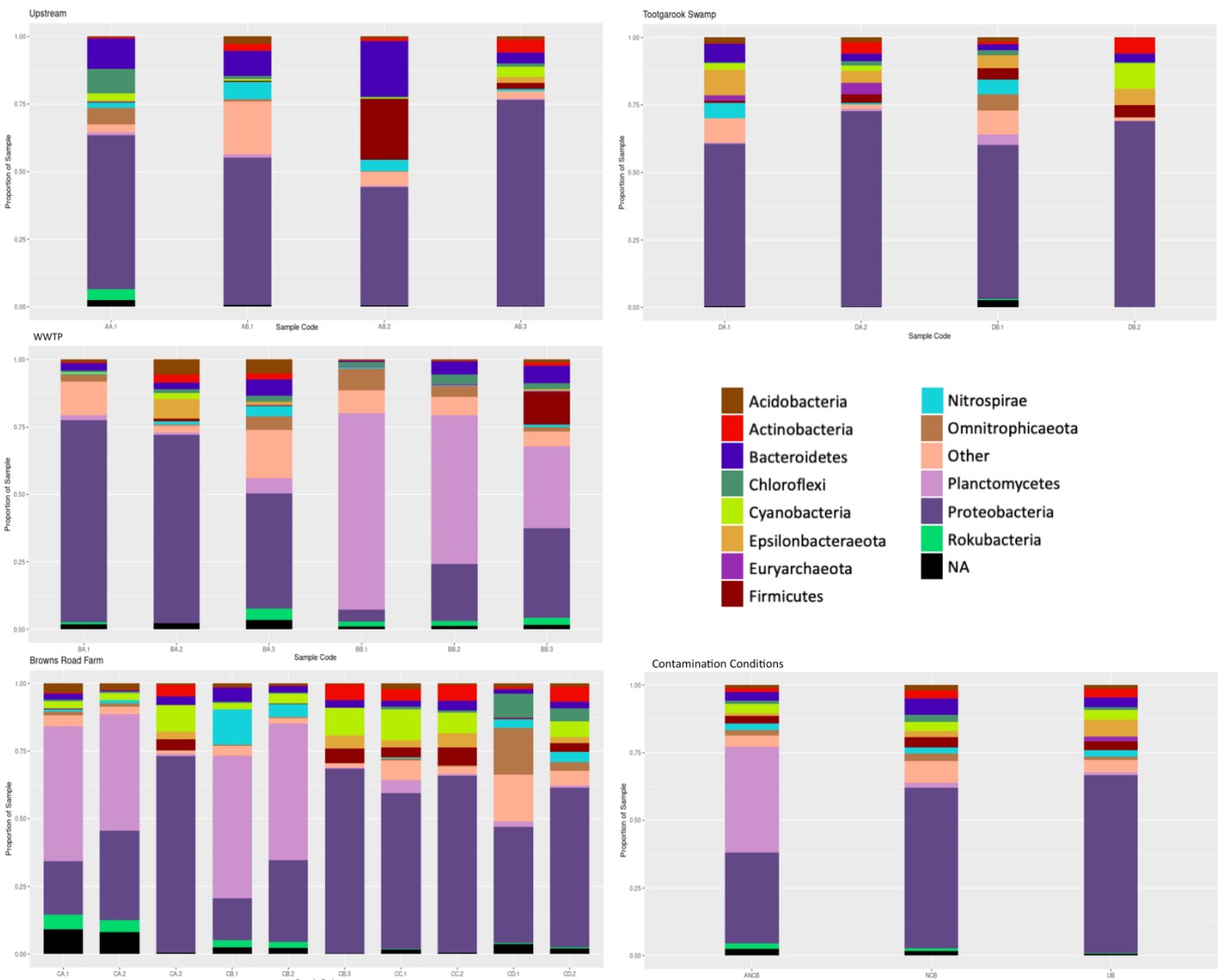

**Figure 2.** Abundances of 13 prominent phyla within the microbial communities of the upstream, wastewater treatment plant, Browns Road Farm and Tootgarook Swamp bore samples. Contaminant conditions shows the averages of the three contaminant conditions: ammonia and nitrate contaminated bores (ANCBs), nitrate contaminated bores (NCBs), uncontaminated bores (UBs). 'Other' represents all identified phyla in the samples that were not included in the legend, NA represents all unidentified phyla in the samples.

The Nitrospirae phylum had an average relative abundance of 2.5% in the UBs, 2.6% in the NCBs, and 2.6% in the ANCBs (Figure 2). The Nitrospirae phylum is metabolically diverse with both aerobic and anaerobic species with mostly chemolithotrophic and mixotrophic metabolisms. Within our samples, Nitrospirae species such as those in the *Nitrospira* genus are known to oxidize nitrite into nitrate and species in the *Leptospirillum* genus are known to oxidize ferrous iron.

The Omnitrophicaeota candidate phylum (formerly OP3) belongs to the PVC superphylum, and had an average relative abundance of 2.4% in the UBs, 4.4% in the NCBs, and 2.3% in the ANCBs (Figure 2). Members of the PVC superphylum are important in carbon and nitrogen cycling, and although there were multiple OTUs identified to the Omnitrophicaeota candidate phylum, only one was identified to genus level (*Candidatus omnitrophus*) in groundwater samples due to Omnitrophicaeota being largely uncharacter-

ized. Omnitrophicaeota is known to inhabit anaerobic environments, and a gene for nitrate reductase has been recently identified in its genome [45,46].

The Cyanobacteria phylum had an average relative abundance of 2.2% in the UBs, 2.1% in the NCBs, and 3.5% in the ANCBs (Figure 2). Some Cyanobacteria can grow as dark heterotrophs at much slower rates than as photoautotrophs, and some cyanobacteria have been reported to synthesize nitrogenase under anaerobic conditions, allowing it to fix $N_2$. None of the Cyanobacteria were identified to the genus level within the analyzed samples, indicating that they may be Cyanobacterial species specifically adapted to groundwater and currently uncharacterized.

Euryarchaeota is a morphologically diverse phylum with an average relative abundance of 2.2% in the UBs, 1.3% in the NCBs, and 1.7% in the ANCBs (Figure 2). Within our samples, the Euryarchaeota phylum was primarily made up of anaerobic methanogens such as the *Methanospirillum* genus [47].

Firmicutes is a metabolically diverse phylum with an average relative abundance of 2.0% in the UBs, 1.3% in the NCBs, and 1.3% in the ANCBs (Figure 2). Across the Firmicutes phylum, most species are chemoorganotrophic species. However, most of the various-trophic metabolisms can be found. Both aerobic and anaerobic species were found; the capability for nitrate reduction in genus such as *Bacillus* and endospore formation in families such as *Bacillaceae* was also detected [48]. In addition, several pathogenic genera were found, such as *Streptococcus* and *Staphylococcus* [49,50].

The Acidobacteria phylum has a primarily chemoorganotrophic metabolism and had an average relative abundance of 1.6% in the UBs, 2.5% in the NCBs, and 1.4% in the ANCBs (Figure 2). The Acidobacteria phylum had a range of both aerobic and anaerobic species [51]. Within the samples, there were genera capable of fermentation, such as the *Holophaga* genus, as well as several species that respire with nitrate within the *Holophagaceae* family [52,53].

The Chloroflexi phylum had an average relative abundance of 1.4% in the UBs, 3.4% in the NCBs, and 1.3% in the ANCBs (Figure 2). Within the samples, the Chloroflexi phylum contained mostly anaerobic species capable of chemotrophic metabolism, such as the *Dehalogenimonas* genus that uses halogenated organics as electron acceptors [54] and the chemoheterotrophic *Anaerolineaceae* family that fermentatively utilize sugars and proteins [55].

The final phylum described shows considerable variation between the three contamination conditions; the Rokubacteria candidate phylum is characterized by the significant genetic diversity shown between individuals, and had an average relative abundance of 0.1% in the UBs, 1.3% in the NCBs and 2.8% in the ANCBs (Figure 2). Rokubacteria have been shown to have a mixotrophic metabolism, and genetic evidence shows the presence of genes such as nitrite oxidoreductases that allow for utilization of various electron donors and acceptors [56]. Within the samples, the *Candidatus Methylomirabilis* genus has been shown to be capable of nitrite-dependent anaerobic methane oxidation.

Interestingly, there were fewer unidentified phyla in the UBs (1.0%), compared to the NCBs (2.6%) and the ANCBs (3.3%) (Figure 2).

### 3.2.3. Constrained Correspondence Analysis

The X axis of the constrained correspondence analysis (CCA) (Figure 3) shows a positive association with $NH_4^+$, Total Kjeldahl nitrogen (TKN), potassium, phosphorus, and redox potential; of these, all except redox potential are highly associated with wastewater [57]. The X axis also exhibits a negative association with DO and bore depth; the negative association with DO was likely due to the oxidation of $NH_4^+$ consuming much of the oxygen in the ammonia contaminated groundwater. Interestingly, DO is typically negatively associated with bore depth due to oxygen naturally becoming scarcer deeper in most aquifer systems. The fact that this was not the case in the current dataset may indicate that the $NH_4^+$ contamination has altered the natural biochemical conditions in the shallow portion of the aquifer (i.e., consuming a significant amount of available DO). The Y axis

was positively associated with $NO_3^-$, organic N, TOC, TDS, Mg, Na, Cl, and $SO_4$. The CCA shown in Figure 3 represents the amalgamation of both the 16S rRNA data and the groundwater chemistry and physicochemical parameter data, to give an overview of the interactions between these two datasets.

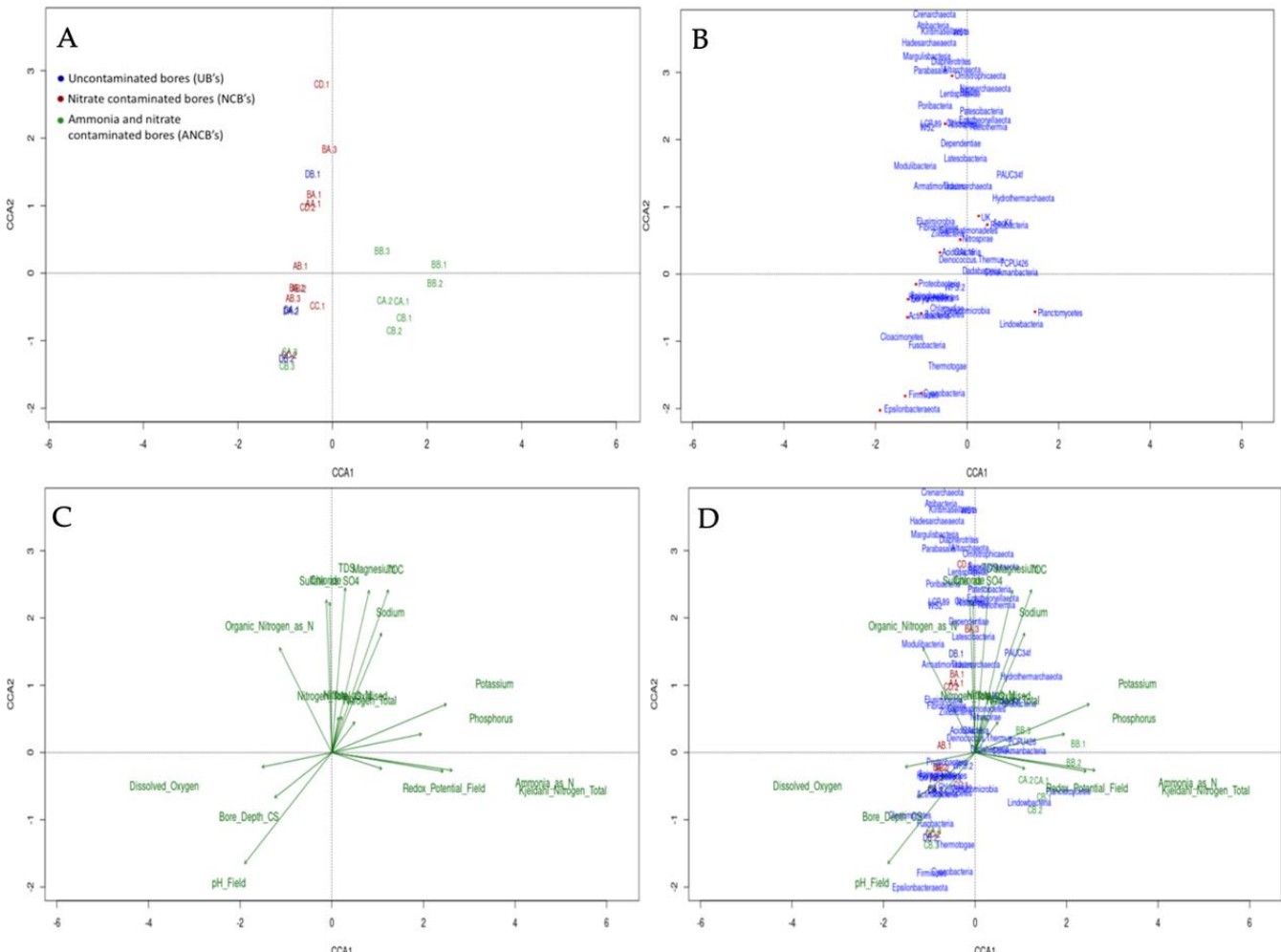

**Figure 3.** Constrained correspondence analysis (CCA) plot showing combined physiochemical and 16S amplicon sequencing data relative to each other compressed into two dimensions. (**A**) samples colored according to their contamination conditions. (**B**) all of the phyla present in the samples, 13 prominent phyla marked with red dots. (**C**) selected Physiochemical vectors. (**D**) all other graphs combined (all graphs are relative to each other). Axis 1 explains 42% of the variance and Axis 2 explains 15% of the variance.

With these axis associations, it was expected that phyla positively associated with $NH_4^+$ would trend towards the right side of the CCA plot, while phyla positively associated with $NO_3^-$ would trend towards the top of the CCA plot; phyla relatively unaffected by either contamination would likely be more centered on the CCA plot. This can be seen in phyla such as Planctomycetes, which occurs in the bottom-right quadrant of the CCA. It should be noted that Planctomycetes was more abundant than the UBs in both the NCBs and ANCBs; the large difference between the Planctomycetes in the NCBs and the ANCBs explains why it was in the bottom right quadrant instead of the top right. Another two examples of these trends are the Epsilonbacteraeota phylum, which was negatively affected by both $NO_3^-$ and $NH_4^+$ contamination and resides in the bottom left quadrant, and the Chloroflexi phylum, which was positively affected by the $NO_3^-$ contamination and slightly

negatively affected by the $NH_4^+$ contamination and resides in the top left quadrant of the CCA (Figure 3).

According to the physicochemical vectors shown in Figure 3 the ANCBs should trend towards the right of the CCA plot, while the NCBs should trend towards the top of the CCA, and the UBs should trend towards the center or bottom left of the CCA plot. Of these three groups, the ANCBs and UBs are both relatively tightly grouped, with few outliers; however, the NCBs are much more spread along the *Y*-axis. This is likely because the large geographic area over which the NCBs samples were collected encompasses multiple different habitat zones—e.g., cattle grazing lands, the WWTP, down-gradient of the market gardens. This may have resulted in differences in the physicochemistry and hence microbial ecology of these different regions, despite the overarching $NO_3^-$ contamination and chemical similarity of the samples. This theory is given credence by the multiple physicochemical properties lining up with the *Y*-axis. There are several aforementioned outliers to these predictions, namely, CA.3, CB.3, CC.1, and DB.1. It is likely that the large rainfall event that occurred in the days leading up to or during the collection of those samples may have affected both the chemical and biological composition of these samples—rainfall in the days leading up to, during and after sampling is shown in Figure 4.

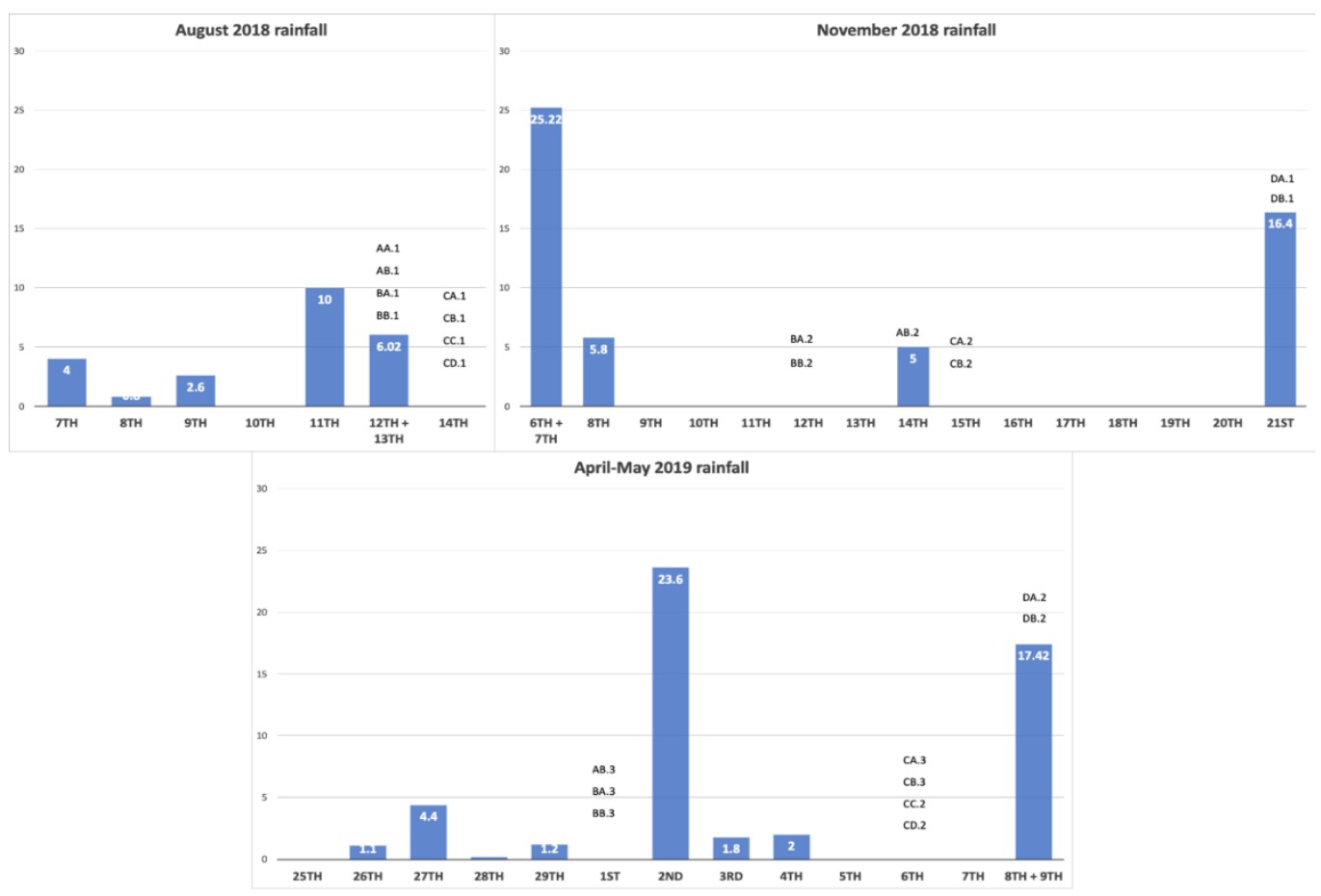

**Figure 4.** Rainfall data for the three sampling rounds. Samples are labelled on the day they were taken. Data were taken for five days before the first sampling event up to the day the last sample was taken. Data obtained from [58].

DSE63273 bore or sample AA.1 falls in the general grouping for the NCBs within Figure 3, but there are differences between it and the other upstream bore RB23 (samples AB.1, AB.2, AB.3); notably, the bore displays both physicochemical and ecological features that are more similar to the Browns Road Farm and WWTP bores than RB23. This leads to the question of which of the upstream bores were more representative of the upstream/regional $NO_3^-$

contamination. DSE63273 was immediately downstream from the Market Gardens, so it was likely the best representation; however, it was also immediately adjacent to a road and a poultry manure stockpile and was likely influenced by these factors [15,59]. Alternately, RB23 was surrounded by a small grove of trees, which also likely impacts the local physio-chemistry and ecology. There may be no single bore that was a perfect representation of the regional $NO_3^-$ contamination, and it is likely that the relevant available bores exist on a spectrum. There were no temporal effects found across the dataset.

3.2.4. Microbial Communities and Their Relationship with Physicochemical Properties

Across the 13 identified phyla and after Bonferroni correction, only Planctomycetes showed a significant interaction with any nitrogen related physicochemical variables within our samples, namely $NH_4^+$. This is unsurprising mainly given the drastic differences in abundance seen between the different contamination conditions. From this, it is clear that Planctomycetes is well adapted to the presence of $NH_4^+$, which likely reflects its ANAMMOX capabilities. Given its drastic changes in abundance in the presence of $NH_4^+$ Planctomycetes could potentially be used as an indicator for the presence of $NH_4^+$. The importance of Planctomycetes as an indicator species may not be immediately obvious given that $NH_4^+$ contamination can be discovered with relative ease through physicochemical testing.

However, the RB17 bore contains no detectable $NH_4^+$, and yet it had a similar abundance of Planctomycetes to RB18, and the other $NH_4^+$ affected bores. If RB17 had a fluctuating concentration of $NH_4^+$ that increased on a regular basis, but was readily consumed by the local ecology, it would be unlikely to be detected by the physicochemical testing and analysis due to the inconsistency of these fluctuations. However, the local ecology would change as a result of the frequent $NH_4^+$ spikes, and the Planctomycetes population may grow enough during the spikes to maintain their population between spikes, reflecting the $NH_4^+$ contamination despite the lack of physicochemical evidence. In this, we see that indicator species (or phyla in this case) can be used to detect fluctuating contamination concentrations that may otherwise go unnoticed. There were also denitrifying species identified in every bore sampled in this project and Adebowale, Surapaneni, Faulkner, McCance, Wang, and Currell [15] also show evidence of denitrification occurring in these bores. However, unlike ANAMMOX species denitrifiers are spread across multiple phyla, and there are different species performing the task across the bores making it difficult to identify an indicator species or even group.

Indicator species can also be utilized to determine the extent of the impact; for example, phyla such as Proteobacteria, Epsilonbacteraeota, and Nitrospirae were not significantly associated with any physicochemical parameters and were also present in all of the samples. This suggests that these phyla are important to the functioning of these ecosystems. They also seem to be resistant to the contaminants present in the samples. Given both their importance and resistance to change, the loss or decline of these phyla would indicate the decline of the ecosystem as a whole.

**4. Conclusions**

In summary $NO_3^-$ contamination seems to increase the overall richness of the affected bores. In contrast, $NH_4^+$ contamination seems to increase the abundance of $NH_4^+$ resistant phyla while negatively impacting non-resistant phyla, resulting in the potential loss of several phyla and lower overall richness.

Planctomycetes showed a significant relationship (*p*-value < 0.01) with the presence of $NH_4^+$. The ANAMMOX bacteria that are characteristic of the Planctomycetes phylum were identified as a potential indicator of $NH_4^+$ contamination. Specifically, *Candidatus Brocadia* could be utilized when physicochemical testing is ineffective, such as in the case of bores that are on the edge of the contaminant plume or are inconsistently contaminated with $NH_4^+$. This shows that characterization of the microbial species can give important insights into biogeochemical processes in contaminated groundwater that could not be gained from standard hydrochemical sampling campaigns.

Future research in this area should focus on more extended studies with more time points, and potentially a site with better positive and negative controls would be ideal. If the analysis of both sequencing and physicochemical data were included in routine testing, our understanding of contaminant and microecological behavior would likely increase significantly.

**Supplementary Materials:** The following supporting information can be downloaded at: https://www.mdpi.com/article/10.3390/w14040613/s1, Figure S1: Abundances of phyla organized by date; Figure S2: CCA organized by date; Table S1: Methods of analysis used by ALS; Table S2: R studio packages used throughout analysis; Table S3: Physiochemical and sample identification data for the samples grouped by sample date; Table S4: Alpha diversity, abundance estimators and evenness measures organized by date; Table S5: P and F values from ANOVA and Tukeys analysis of alpha diversity, evenness and abundance estimators

**Author Contributions:** Conceptualization, methodology, software, formal analysis, investigation, data curation and writing—original draft preparation was all completed by J.G.M. Supervision and project administration were the responsibilities of A.S.B., M.J.C., S.M.R., A.S. and M.M. Funding acquisition: A.S.B., M.J.C., S.M.R., A.S., M.M., N.D.C., S.A. and W.R. Writing—review and editing was completed by A.S.B., M.J.C., S.M.R., A.S., M.M., N.D.C., S.A., D.H., W.R. and J.G.M. Comments and edits were organised and enacted by J.G.M. All authors have read and agreed to the published version of the manuscript.

**Funding:** This research was funded by CRC CARE with in kind support from Gippsland Water, Melbourne Water, South East Water, Western Water (Grant No. 2.3.02).

**Data Availability Statement:** 16S microbial community data and environmental data can be found in the NCBI Sequence Read Archive (SRA) (https://trace.ncbi.nlm.nih.gov/Traces/sra/sra.cgi?view=studies (accessed on 10 February 2022)) under the heading 'Groundwater microbial communities in an unconfined nitrogen contaminated aquifer' and will be available after 1 July 2022.

**Acknowledgments:** The authors would like to thank CRC CARE, Gippsland Water, Melbourne Water, South East Water, Western Water, and BlueSphere Environmental for their funding and support in undertaking this work (Grant No. 2.3.02). Additionally, we would like to thank Esmaeil Shahsavari for his generous assistance with the sequencing, and Kirill Tsyganov and the Monash Bioinformatics Platform for their assistance with the analysis of the microbial community data.

**Conflicts of Interest:** The authors declare no conflict of interest.

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
