# Peer review of "The Variation in Groundwater Microbial Communities in an Unconfined Aquifer Contaminated by Multiple Nitrogen Contamination Sources"

_water, doi:10.3390/w14040613_

Round 1

Reviewer 1 Report

General comments

I enjoyed reviewing the manuscript. The only possibly major comment is on the lack of temporal trends from three sampling events across 9 month period. Please see more for the specific comments below.

Specific comments

Reference [1] – Remove it since it was never cited in the text.

L132-133: Were results processed and analyzed by any means to reveal the temporal variability? Or were the measurements fairly consistent throughout sampling? Please provide measurements and microbial communities per sampling events in supplementary results to supplement Tables 1 and 2, and Figures 2 and 3.

L268-271: Why is the case? WWTPs are designed to oxidize ammonia (as well as organic C) aerobically, so this is unexpected. I would like to see some explanation about this finding.

L357: I would like to see the statistical test results among three contaminant conditions on diversity measures qualitatively presented in the current section.

L383: Please present data or other evidence on these invasive taxa, which is found here but not in ANCB samples.

L389-390: These are relative abundances. Is the trend the same in absolute abundance?

L393-394: How do the abundances of these AMO taxa change between contaminant conditions?

L411-412: The abundance of Planctomycetes indicates an anoxic condition, then why didn’t the denitrifying populations and processes were detected?

L415: ANAMMOX simultaneously oxidizes and reduces ammonia and nitrite, not nitrate.

Figure 2. Please explain the large difference in microbial community composition in CA3 and CB3 where Planctomycetes are disappeared.

Figure 3. I want to make one statement regarding CCA here due to an overall lack of confidence with these results. Panel C indicates the existence of collinearity, so please take care of those constraining variables using VIF. Perhaps one more thing: report variances explained with the first two axes here.

L613-615: I agree with the authors here, but phyla are not “species”. Update here or the following paragraphs to be compatible.

Conclusion: The current conclusion is more like a summary of results & discussion. I would like to see a concise and impactful conclusion here.

Reviewer 2 Report

This manuscript presents an important analysis of microbial communities in groundwater. I think results would benefit from application of differential abundance analysis, like ANCOM-BC (www.nature.com/articles/s41467-020-17041-7,) to test the hypothesis that particular ribotypes are positively or negatively associated with environmental parameters. Style and format edits are addressed in the attached pdf.

Author Response

Response to reviewer #2’s comments:

This manuscript presents an important analysis of microbial communities in groundwater. I think results would benefit from application of differential abundance analysis, like ANCOM-BC (www.nature.com/articles/s41467-020-17041-7,) to test the hypothesis that particular ribotypes are positively or negatively associated with environmental parameters.

Response: This seems to be a very robust form of analysis; however, this paper is aimed to show the trends and indicators that can be found with relatively simple analysis. Additionally, after reading the ANCOM-BC paper I am unsure that our data fits all the assumptions for the test. We have performed some additional statistical analysis in this edit which can be seen at lines 369-387. 

Responses to the comments from reviewer 2 in the PDF have been addressed in the PDF supplied by reviewer 2. As there were many typographical and grammatical adjustments, we have left only the unchanged comments and edits with an explanation of why commented below for convenience.

Reviewer 3 Report

why were the tests on the four seasons not carried out as is usually done?

Round 2

Reviewer 1 Report

Figure 3.1. 

The blog posting listed is appropriate that vif doesn’t tell which collinear variable is more important than others, which is determined by the researcher with knowledge. It is not limited problem here, rather common problems for ones to put statistics over biology. I still think it is better to remove those collinear variables with more representing (statistically and biologically) variable(s) for the sake of CCA model relevancy, but I can also buy the authors argument for the collinear nitrogen variables may be informative.
